# Platelet Functions During Extracorporeal Membrane Oxygenation. Platelet–Leukocyte Aggregates Analyzed by Flow Cytometry as a Promising Tool to Monitor Platelet Activation

**DOI:** 10.3390/jcm9082361

**Published:** 2020-07-23

**Authors:** Alexandre Mansour, Mikael Roussel, Pascale Gaussem, Fabienne Nédelec-Gac, Adeline Pontis, Erwan Flécher, Christilla Bachelot-Loza, Isabelle Gouin-Thibault

**Affiliations:** 1Department of Anesthesiology Critical Care Medicine and Perioperative Medicine, Rennes University Hospital, F-35000 Rennes, France; alexandre.mansour@chu-rennes.fr; 2Rennes University Hospital, INSERM-CIC 1414, F-35000 Rennes, France; 3Innovative Therapies in Haemostasis, Paris University, INSERM U1140, F-75006 Paris, France; pascale.gaussem@aphp.fr (P.G.); christilla.bachelot-loza@parisdescartes.fr (C.B.-L.); 4Department of Biological Hematology, Rennes University Hospital, F-35000 Rennes, France; mikael.roussel@chu-rennes.fr (M.R.); Fabienne.NEDELEC.GAC@chu-rennes.fr (F.N.-G.); adeline.pontis@chu-rennes.fr (A.P.); 5Microenvironment, Cell Differentiation, Immunology and Cancer, Rennes University, INSERM U1236, F-35000 Rennes, France; 6Cytometrie Hematologique Francophone Association (CytHem), F-75013 Paris, France; 7Department of Biological Hematology, AH-HP, Georges Pompidou European University Hospital, F-75015 Paris, France; 8Cardio-Thoracic Surgery, Rennes University Hospital, INSERM U1099, F-35000 Rennes, France; erwan.flecher@chu-rennes.fr

**Keywords:** platelet activation, platelet aggregation, von Willebrand factor, leukocytes, platelet-leukocyte aggregates

## Abstract

Extracorporeal membrane oxygenation (ECMO) is an extracorporeal circulation used to manage patients with severe circulatory or respiratory failure. It is associated with both high bleeding and thrombosis risks, mainly as a result of biomaterial/blood interface phenomena, high shear stress, and complex inflammatory response involving the activation of coagulation and complement systems, endothelial cells, leukocytes, and platelets. Besides their critical role in hemostasis, platelets are important players in inflammatory reactions, especially due to their ability to bind and activate leukocytes. Hence, we reviewed studies on platelet function of ECMO patients. Moreover, we addressed the issue of platelet–leukocyte aggregates (PLAs), which is a key step in both platelet and leukocyte activation, and deserves to be investigated in these patients. A reduced expression of GPIb and GPVI was found under ECMO therapy, due to the shedding processes. However, defective platelet aggregation is inconsistently reported and is still not clearly defined. Due to the high susceptibility of PLAs to pre-analytical conditions, defining and strictly adhering to a rigorous laboratory methodology is essential for reliable and reproducible results, especially in the setting of complex inflammatory situations like ECMO. We provide results on sample preparation and flow cytometric whole blood evaluation of circulating PLAs.

## 1. Introduction

Extracorporeal circulation is used to manage patients with severe organ dysfunction. Extracorporeal membrane oxygenation (ECMO) provides gas exchange during severe respiratory failure with veno-venous (VV-ECMO) circuits [1]. Veno-arterial ECMO (VA-ECMO), also called extracorporeal life support (ECLS), has been increasingly used in life-threatening circulatory failure refractory in conventional treatments, with the aim of maintaining satisfactory tissue perfusion pending myocardial recovery or heart transplantation. In all these devices, blood comes into sustained contact with a large artificial material surface area and a pump, leading to a complex inflammatory response involving activation of the coagulation system, complement system, endothelial cells, leukocytes, and platelets [1]. The pump can be either centrifugal or pulsatile with different rotational speeds and variable effects on hemostasis and bleeding [2]. ECMO is associated with bleeding and thrombotic complications, which remain a leading cause of morbidity and mortality of such fragile patients. Therefore, managing hemostasis to avoid clot formation without increasing bleeding risk in patients with ECMO, remains a major challenge. According to the recent review by Doyle et al. the rate of ECMO-associated venous thromboembolism (VTE) widely varies from 18% to 85%, depending on the studies and the anticoagulation regimens, with oxygenator clotting thrombosis in around 10–16% of patients, depending on the circuit type and patient age [1]. Severe hemorrhage is reported in nearly 40% of patients, with intracranial hemorrhage in 16–21% of them [1]. The pathogenesis of these complications, which can be related to the underlying disease or to the ECMO circuit, is therefore multifactorial and is not fully elucidated. Vessel damage and low blood flow at cannula sites, immobility, disseminated intravascular coagulation, elevated factor VIII, procoagulant hydrophobic artificial surfaces, hemolysis, circulating extracellular vesicles, and monocyte tissue factor (TF) expression, all contribute to the prothrombotic changes in patients receiving ECMO [1]. Beyond TF-based activation of coagulation, contact activation through artificial surfaces involving notably FXII and FXI, might also play an important role in clot formation.

Concerning the bleeding risk, acquired von Willebrand syndrome (AVWS) was reported in ECMO patients [3,4,5]. AVWS is characterized by a defect in the multimeric von Willebrand factor composition, which is a key factor during primary hemostasis. Upon exposition onto artificial surfaces and high shear stress, von Willebrand factor unfolds, which facilitates subsequent proteolysis and cleavage of large multimers by the ADAMTS-13 protease. The loss of the high-molecular weight multimers results in a decreased binding of von Willebrand factor to collagen and platelets. AVWS rapidly occurs, within 24 h after ECMO implantation, and is reversed few hours after explantation [3]. Thrombocytopenia, which is frequently observed in these patients, with a median platelet count between 50 and 100 × 10^9^/L (see below), might contribute to the hemorrhagic risk. It was shown that ECMO patients who experienced bleeding events had a significantly lower platelet nadir than patients without bleeding [6]. A platelet dysfunction was also reported but not clearly established, with some studies showing platelet glycoprotein (GP) Ibα and GPIV shedding, as detailed below.

In addition to promoting thrombus, platelets are involved in inflammatory reactions by their ability to release stored and newly formed mediators. They also interact with leukocytes and play an essential role for leukocyte recruitment and initiation and propagation of inflammation. The purpose of this review was, therefore, to summarize the various roles of platelets in the context of ECMO in adult patients, with a special focus on the potential usefulness of studying platelet–leukocyte interactions, by measuring the so-called platelet–leukocyte aggregates (PLAs) through flow cytometry.

## 2. Platelet Activation, Aggregation, and ECMO

In a systematic recent review, Balle et al. reported that there are only a limited number of published studies investigating platelet functions in patients during ECMO therapy. These studies suggest a reduced potential for platelet adhesion, activation, and aggregation during ECMO [7].

Shear stress plays an essential role in regulating platelet activation and aggregation through VWF (von Willebrand Factor) binding to platelet GPIb within the platelet GPIb–IX–V complex, with limited binding under the low shear condition. Above a critical shear rate the structural change of VWF enables its binding to GPIbα with increased affinity. This GPIbα–VWF interaction, leading to platelet translocation and rolling, enables stable platelet adhesion on collagen and subsequent GPVI-mediated activation of α_IIb_β_3_ and other platelet integrins [8]. GPIbα and GPVI shedding occurs via activation of the constitutively present platelet membrane-bound ADAM-17 and ADAM-10 metalloproteases, and as a consequence downregulates platelet adhesion and signaling [8]. The activation mechanism of ADAM-17 and ADAM-10 is not clearly defined but the trans-membrane protein tetraspanins and iRhoms could be involved in the regulation of their activity [9]. Whether high shear rate leads to the direct activation of ADAM metalloproteases and triggers platelet receptor shedding, needs to be further investigated.

In ECMO, shear-induced shedding might not require platelet signaling pathways or activation of α_IIb_β_3_ and could be a direct effect of exposure to fluid shear stress [9]. The analysis of the level of adhesion/activation receptors on circulating platelets in patients receiving ventricular assist devices or ECMO therapy were first reported by Lukito et al. in 2016 [10]. This single-center observational study enrolled 20 patients (14 VA-ECMO and 6 VV-ECMO, Medtronic centrifugal pumps), with a median time post-ECMO initiation of four days (range: 1–11 days). The authors reported that platelet receptor shedding was demonstrated by a significantly reduced surface GPIbα and GPVI levels, and a concomitant increase in plasma soluble GPVI in patients, compared to healthy donors. However, since the level of glycocalicin, which is a large proteolytic extracellular fragment of the GPIbαreceptor released upon GPIbα proteolysis was not measured, shedding of GPIbα could not be distinguished from internalization. By contrast, there was no significant loss of _IIb_ subunit of integrin α_IIb_β_3_ (also named GPIIb/IIIa), which is a platelet-specific receptor subunit that is not susceptible to proteolysis [10].

In another cohort of 20 VA-ECMO patients who had a mean platelet count of three time-points comprised between 84 × 10^9^/L to 108 × 10^9^/L, the authors demonstrated a severely reduced platelet adhesion and aggregation, under a high shear rate of 2000 s^−1^, in a flow chamber with collagen-coated channels. Adhesion and aggregation improved but did not normalize with in vitro addition of VWF concentrate. The circulating glycocalicin level was found to be elevated, consistent with an increased GPIb proteolysis [11].

Concerning platelet aggregation, most published studies used whole blood impedance aggregometry with the Multiplate^®^ analyzer and adenosine diphosphate (ADP), thrombin-receptor-agonist-peptide-6 (TRAP), and arachidonic acid (AA) as activators, or ristocetin (Table 1).

Multiplate^®^ tests were performed along with thromboelastometry (ROTEM^®^) on citrated whole blood of seven patients, over a 110 day-period. Thrombocytopenia was found in 76% of patients. Thromboelastometry maximum clot firmness (MCF), which is very sensitive to fibrinogen and to a lesser extent to platelet count, was highly specific, but not very sensitive in the prediction of bleeding (MCF-intrinsic rotational thromboelastometry, specificity: 91% and sensitivity: 53%). Unfortunately, details on the timing of the test and the bleeding events were lacking. Aggregation tests were performed only in samples with a platelet count above 100 × 10^9^/L. They were inconsistently abnormal with low platelet function in 72% of tests performed with ADP, 50% with collagen and TRAP, and 61% with ristocetin [12].

Platelet aggregation in hirudin anticoagulated whole blood of 38 patients, activated with TRAP, ADP or AA, and analyzed with the Multiplate^®^ analyzer, decreased by 30% to 40%, after 24 h on ECMO, compared to baseline, and was significantly associated with transfusion requirement. After 48 h on ECMO, aggregation induced by TRAP did not differ from baseline, while those induced by ADP and AA were still lower than baseline values. Within 24 h of weaning from ECMO, the platelet count and function returned to the baseline values [13].

In a recent study, platelet aggregometry was assessed on whole blood samples of 20 patients collected in heparin calcium-balanced tubes, provided that platelet count was ≥ 70 × 10^9^/L. It should be mentioned that median aggregation results using AA, ADP, and TRAP as activators were below the normal reference range given by the manufacturer, before VV-ECMO initiation. Platelet count significantly dropped six hours after implementation of ECMO therapy to a median level of 126 × 10^9^/L and continuously decreased thereafter with a median level of 102 × 10^9^/L on day 4. Platelet aggregation was significantly reduced six hours after VV-ECMO initiation, compared to before, and spontaneously recovered and exceeded baseline values on day 2 and after, in contrast to the continuously decreasing platelet counts. There was no difference in platelet aggregation results between patients with and without signs of bleeding. As discussed by the authors, the recovery of platelet aggregation after day two, might be due to a decreasing effect of factors like the initial hemodilution under extracorporeal circulation [14].

Finally, only in one study, platelet count was taken into consideration and results were compared with the 95% prediction interval of platelet aggregation obtained in healthy whole blood at various platelet counts. Platelet activation was also measured by flow cytometry as the increase in expression of platelet surface P-selectin (CD62P, alpha granule membrane protein) and CD63 (lysosome and dense granule membrane protein), following activation, as well as bound fibrinogen to α_IIb_β_3_. Hirudin anticoagulated whole blood was used for impedance aggregometry analyses, and citrated whole blood for flow cytometry analysis. Following ECMO initiation, platelet count significantly decreased and remained low during ECMO therapy, despite transfusion of platelet concentrates to 20 patients to maintain platelet count above 50 × 10^9^/L. The findings indicated that platelet aggregation assessed with the Multiplate^®^ during ECMO therapy was not impaired when interpreted relative to platelet count. When using flow cytometry, no association was found between platelet count and activation. With this method less dependent on platelet count, they found that platelets demonstrated a reduced ability to become activated. The authors suggested that this could actually be consecutive to the vast activation of platelets in the ECMO circuit, related to the high shear stress, consistent with the increased surface expression of the activated fibrinogen receptor at rest, and a decreased expression of P-selectin, probably due to its shedding [15]. This was in agreement with previous data from our group in the setting of cardiopulmonary bypass in children [16].

In the study of Kalbhenn et al., the reference method, i.e., light transmission aggregometry, was performed in a subgroup of patients with VV-ECMO treatment. All patients had a platelet count > 100 × 10^9^/L and platelet-rich plasma count was adjusted to 250 × 10^9^/L, however, contrary to what is currently recommended. After stimulation with ADP, collagen, and epinephrine, hypoaggregability and altered agglutination induced with ristocetin were found, with incomplete recovery of platelet function on day 3 after explantation. Flow-cytometric platelet analysis revealed severely reduced expression of CD62P and CD63 during ECMO (impaired and dense granules secretion), when platelets were stimulated with various thrombin concentrations, as compared to healthy subjects, while CD41 (α_IIb_), CD42a (GPIX), and CD42b (GPIb) expressions, as well as ADP-induced fibrinogen-binding were normal [17].

To summarize, concerning the effect of ECMO on platelet adhesion receptors and membrane proteins, discrepancies between studies exist. Some studies report a reduced level of GPIb and GPVI on circulating platelets that were exposed to abnormal shears. This finding supports the hypothesis that the association of a reduced level of platelet surface receptors with thrombocytopenia and AVWS could contribute to bleeding events in this situation of elevated shear rates. However, the design of these studies and the different preanalytical conditions cannot allow drawing firm conclusions on the effect of ECMO therapy on platelet function. Moreover, the type of device as well as their rotational speed was sparsely reported, while they could have different effects on platelets. To add to the complexity, patients who require VA-ECMO or VV-ECMO, have different underlying diseases with various resulting effects on hemostasis. Studies specifically taking into account these variables would be informative. Moreover, the studies were not powered to establish relationships between bleeding events and platelet dysfunction. Concerning platelet aggregation, abnormalities were inconsistently reported. The use of multiple electrode aggregometry analyzer in most studies presents the advantage to assess platelet function in whole blood, but does not allow to specifically investigate platelet dysfunction per se. While low hematocrit is not thought to have a major influence on Multiplate^®^ analyzer results, platelet count ≤ 100 × 10^9^/L, which is frequently encountered in patients with ECMO treatment, leads to reduced platelet aggregation with Multiplate^®^ analyzer system, and therefore, could render the interpretation of the findings to be complicated. Platelet count should, therefore, be taken into account in analyzing the results of the studies (Table 1).

## 3. Inflammation in ECMO Patients

One of the relevant complications of ECMO is the inflammatory response. Following ECMO initiation, the association of an increase in pro-inflammatory cytokines with immune system-activation might contribute to end-organ dysfunction and death. However, as with bleeding and thrombotic complications, it is difficult to evaluate the respective part of the ECMO circuit per se in the inflammatory response and that of the underlying disease and its management. Unlike the endothelium, artificial surfaces have no regulatory molecules of the activation of the complement system. Hence, artificial surfaces contribute to the propagation and positive feedback of the complement cascade, leading to an excessive inflammatory response and capillary leak syndrome, which is a complication of extracorporeal circuits [1,18]. Following ECMO initiation, the contact system of coagulation becomes also activated and activated FXII (FXIIa) triggers the kallikrein–kinin pathway leading to the release of the proinflammatory peptide hormone bradykinin. In line with this, molecules inhibiting FXIIa were shown to reduce inflammation in ECMO models [19].

## 4. Inflammation and Platelet–Leukocyte Interactions

It is likely that the platelet-related inflammatory and immune functions that are triggered in host defense and in inflammatory syndromes or diseases are also set off in ECMO patients. Hemostasis and inflammation are intimately linked, and induce and amplify one another with beneficial or deleterious consequences, depending on the clinical situation [20,21]. The different mechanisms sustaining the interaction between platelets and inflammation were extensively described in recent reviews [20,21]. In brief, platelets recognize and respond to pathogens through the expression of multiple Toll-like receptors and other receptor classes that mediate inflammatory and immune signaling. Platelets are involved in endothelial barrier function and vascular permeability, through various cell adhesion molecules acting at the interface between platelets and endothelial cells. Upon activation, the platelet release stored inflammatory mediators and immunomodulators (RANTES, CD40, Platelet Factor 4 (PF4), Neutrophil-Activating Peptide-2, Platelet Activating Factor, Transforming Growth Factor-…). They synthesize reactive oxygen species, inflammatory proteins (Interleukin-1β…), and produce extracellular vesicles with inflammatory, immune, and thrombogenic activities. They induce neutrophil extracellular traps (NETs) and interact with leukocytes forming PLAs, thereby inducing new gene expression and synthesis of inflammatory mediators in those cells [20,22].

Different subsets of leukocytes can interact with platelets to form PLAs, leading to various effects. Among all leukocytes, monocytes show the highest affinity for platelet P-selectin, followed by neutrophils; lymphocytes have the lowest affinity [23]. It was shown that among monocytes, different subclasses are involved in PLA, in normal subjects and in patients with myocardial infarction as well [24]. The formation of PLAs involves the release of mediators and mutual activation of both cell types. Platelets are the predominant source of both P-selectin and CD40L, which are initially located in the membrane of α granules, and become expressed on the plasma membrane upon platelet activation, enabling the interaction between platelets and leukocytes, which is a critical step in platelet-mediated inflammation. P-selectin cross-links platelets and leukocytes through its corresponding ligand P-selectin glycoprotein ligand-1 (PSGL-1), expressed on the surface of neutrophils, monocytes, dendritic cells, subclasses of lymphocytes, and endothelial cells [21]. Platelet P-selectin binding to PSGL-1 promotes the activation of Mac-1 (α_M_β_2_) and LFA-1 (α_L_β_2_) on neutrophils and β_1_ and β_2_ integrins on monocytes and lymphocytes. The main role of Mac-1 is then to mediate the firm adhesion of neutrophils, monocytes and of some subclasses of lymphocytes to activated platelets and, therefore, to stabilize PLAs [25] (Figure 1). While the resting platelets do not interact with resting leukocytes, they are able to interplay with activated leukocytes, through the interaction of platelet GPIbα with the activated Mac-1. Integrins expressed on platelet plasma membrane also play a role in the interaction with the subendothelial, extracellular matrix, leukocytes, and endothelial cells, and are capable of transducing activating signals [22]. The _IIb3_ integrin is crucially involved in the initiation and regulation of interactions of platelets with leukocytes, under inflammatory conditions. Platelet _IIb3_ serves as a binding partner of the integrin Mac-1 via a fibrinogen bridge, which initiates outside-in signaling into leukocytes and is necessary for leukocyte recruitment, NET formation, and ROS production [26]. Platelets CD40L also interact with CD40 on endothelial cells to promote secretion of chemokines and expression of adhesion molecules [26]. This facilitates migration of leukocytes to the site of vascular injury and subsequent adhesion [27]. Platelet–leukocyte interaction has a functionally important role in leukocyte rolling and adhesion to platelets and endothelium, which are critical steps in the process of transendothelial migration, leukocyte extravasation to the site of inflammation and initiation of vascular inflammation [27,28,29]. During the interaction of platelets with monocytes, expression of tissue factor by the latter initiates coagulation and contributes to thrombus formation. Moreover, after recruitment to the vessel wall, monocytes extravasate into the surrounding tissues and differentiate into macrophages [23].

The effect of platelet–leukocyte interplays on the modulation of monocyte, neutrophil, and lymphocyte activation and function, and the subsequent effect of the different types of PLAs in inflammatory disorders were recently thoroughly reviewed [23,30]. While the prominent role of platelets in pulmonary neutrophil migration and in the recruitment and activation of neutrophils was specifically shown during the pathogenesis of acute lung injury in other settings than ECMO [22,31,32], neutrophil infiltration was described to lead to ECMO-associated lung injury and end-organ damage [19].

Due to the involvement of platelets in the inflammatory process, platelet inhibition was evaluated in some studies to reduce sepsis-associated inflammation. As a receptor of ADP, an agonist originating mainly from dense granule secretion, the platelet P2Y_12_ receptor has a central role in the amplification of platelet activation, in response to a number of agonists. P2Y_12_ is coupled to the Gi protein, the activation of which leads to platelet aggregation, potentiation of granule release, procoagulant activity, and PLA formation. By inhibiting platelet reactivity to ADP and a broad range of other agonists, P2Y_12_ inhibitors reduce the release of pro-inflammatory mediators by platelets. The inhibition of platelet P2Y_12_-mediated platelet-leukocyte interactions is thought to be one of the main mechanisms through which P2Y_12_ inhibitors affect inflammation [28,33]. Using a mouse model of intra-abdominal sepsis and acute lung injury, it was shown that inhibiting P2Y_12_ led to a decrease in platelet activation, platelet–leukocyte interactions, and lung injury, suggesting a key role for activated platelets and the P2Y_12_ receptor during sepsis [34,35]. In the PLATelet inhibition and patient Outcomes (PLATO) trial comparing ticagrelor with clopidogrel, in patients with acute coronary syndromes, there were fewer deaths attributed to sepsis in the ticagrelor group. The more potent anti-P2Y_12_ effect of ticagrelor, combined with its inhibition of adenosine reuptake by red blood cells, might account for the difference in clopidogrel and ticagrelor in this trial [36]. Indeed, adenosine is formed from ATP and ADP released locally at the sites of ischemia, tissue damage, and inflammation with a very short half-life in circulation, due to its rapid cellular uptake and intracellular metabolism. Ticagrelor inhibits the ENT1 transporter and thereby reduces the cellular uptake of adenosine, resulting in increased adenosine-induced responses such as platelet inhibition and neutrophil chemotaxis and phagocytosis [37,38].

The investigation of the interaction between leukocytes and platelets was facilitated by the use of flow cytometric analysis and advanced microscopy techniques [30]. However, characteristics of PLAs in terms of size and number of platelets bound per leukocytes are not well defined. In vitro experiments reported that after platelet activation with thrombin, the semi-quantitative estimate of the number of platelets bound per leukocyte was 2 and 22 for inactivated and activated platelets, respectively [39]. Images showing PLAs as investigated by flow cytometry, tissue section or live cell imaging were presented by Finsterbush et al. in their review on platelet–leukocyte interactions in acute ischemic stroke, renal diseases, and hepatic as well as lung inflammation and infection. They suggested that PLAs within the circulation or locally at the sites of inflammation might represent markers of many thrombo-inflammatory diseases and could be used for the assessment of both thrombotic risk and disease progression [30]. Moreover, PLAs could be of great interest as markers of the modulation of thrombo-inflammation in patients receiving antithrombotic drugs in the setting of acute inflammation.

Despite the potential implication of platelets in inflammation and hemostasis complications of ECMO, to the best of our knowledge, no data are currently available on circulating PLA levels in these patients. Just one recent study was performed on membrane oxygenators taken from 21 patients with VV-ECMO. Membrane oxygenators were collected after termination of ECMO therapy or after replacement during therapy. They were found to be loaded with VWF, activated platelets and leukocytes, and co-localization of nucleated cells (DAPI-positive) and P-selectin-positive structures, consistent with locally formed PLAs. The clinical relevance of these results needs further investigation [40].

At present, much of what is known about PLAs and artificial devices comes from studies in patients undergoing cardiac surgery under cardiopulmonary bypass, but the effects of short-term extracorporeal circulation on platelet function might not be fully translatable to long-term extracorporeal circulation such as in ECMO therapy. Li et al. showed that coronary artery bypass grafting (CABG) induced marked pro-thrombotic and inflammatory responses, which persisted for at least one week. Platelet activation, platelet reactivity, PLA formation, thrombin generation, and TNF-release showed a second peak one week after surgery. These findings suggested that intensified and prolonged antithrombotic/inflammatory treatment should be considered after CABG surgery [41]. In a recent work from the same team, platelet-monocyte and platelet–neutrophil levels were lower three months after CABG than before, suggesting an improvement in wound healing and inflammation, whereas signs of mildly increased platelet reactivity persisted up to three months after surgery. The authors concluded that their data supported the need for efficient antiplatelet therapy during the first three month after CABG [29].

As pointed out by Finsterbush et al. in their review of the different methods to measure PLAs, to allow validating PLAs as a biomarker in the future, protocols for standardized sample preparation and robust reference values need to be established first. Histochemical and immunofluorescent imaging, coupled with advanced confocal or electron microscopy, provide high-resolution images to visualize platelet–leukocyte interactions within tissues [30]. Finsterbush et al. also described different methods to visualize PLAs in vivo—intravital microscopy, which allows tracking cells in live animals over longer time periods in organs; intravenous administration of fluorochrome-conjugated monoclonal antibodies, and microfluidic assays to analyze platelet–leukocyte interaction dynamics under well-defined conditions, including investigation of human cells [30]. Nevertheless, flow cytometry remains the method of choice for measuring circulating PLAs, representing a reliable method that enables concomitant analysis of platelet activation and PLA formation, within the same blood sample [30]. However, the relationship between tissue or in vivo PLAs and circulating PLAs measured ex-vivo through flow cytometry is not yet established.

Moreover, the measurement of PLAs requires specific conditions to avoid ex vivo platelet activation and subsequent neoformation of PLAs; pre-analytical conditions need to be strictly defined and standardization of data analysis are also required.

## 5. Flow Cytometric Whole Blood Measurement of Platelet–Leukocyte Aggregates

### 5.1. General Considerations

Flow cytometry allows rapid and sensitive analysis of fluorescent-labeled cells passing through a flow cell and hit by a focused beam of a laser light. Detectors analyze the light scattering and the emitted fluorescence of each cell or aggregate. Forward-scattered light (FS) and side-scattered light (SS) give physical information regarding cell dimensions and structural complexity. Fluorescence measurement of fluorophore-conjugated monoclonal antibodies allows the analysis of presence and cell density of specific targeted cell antigens, hence permitting study of cell phenotype, function, and interaction with other cells [42].

PLA measurement with whole blood flow cytometry requires at least two fluorescent-labeled antibodies, targeting platelets and leukocytes, respectively. PLAs are then expressed as a percentage of total leukocytes positive for a platelet-specific marker [41,43]. Hence, being independent of absolute leukocyte number and considering the abundance of circulating platelets over leukocytes, PLAs can be measured in both leukopenic and thrombocytopenic patients.

Leukocytes are usually labeled using anti-CD45 (leukocyte common antigen) pan-leukocyte antibody and platelet-specific labeling that typically includes CD41 (GPIIb, α_IIb_), the non-platelet specific CD61 (GPIIIa, β_3_), CD42a, or CD42b. Among these, the platelet α_IIb_β_3_ integrin is often used as the platelet-specific antibody target because of the high copy number of this receptor, i.e., a mean of around 50,000 copies per platelet at rest. Similarly, as specific platelet glycoproteins, CD42a and CD42b are potential targets but should be used with caution, as platelet activation downregulates their expression [44].

Upon activation, platelet extracellular vesicles, measuring from 100 to 1000 nm, are released and circulated, and it cannot be ruled out that they form complexes with leukocytes [24]. Since these complexes are CD41-positive, they can hardly be distinguished from PLAs with platelets. However, both result from platelet activation and reflect interactions with leukocytes.

It should always be kept in mind that PLA formation is a dynamic process and that formation and dissociation of aggregates can also occur in vitro. Therefore, potential influence of platelet-specific antibodies on PLAs in vitro must be considered, especially using CD42b, giving its role in PLA stabilization [45].

Neutrophils, monocytes, and lymphocytes can be distinguished, based on light-scattering properties, allowing basic discrimination of PLA subtypes (platelet–neutrophil, platelet–monocyte, and platelet–lymphocyte aggregates) [30]. Platelet–platelet aggregates that might be present in samples can also interfere with light-scatter gating. Thus, identification of PLA subtypes, while avoiding detection of platelet–platelet aggregates, is improved by using leukocyte subtype-specific antibodies (e.g., CD66b for neutrophils; CD14 for monocytes; CD3, CD4, and CD8 for T-cells; CD19 for B-cells; CD56 for NK-cells) [44].

### 5.2. Pre-Analytical Requirements—Choice of Anticoagulant, Delay between Sampling and Immunolabeling, Sample Stability after Immunolabeling, Effect of Strong Vortex Agitation prior to Immunolabeling

Given the high sensitivity of the flow cytometry assay and the high reactivity of human platelets, PLA measurement is extremely susceptible to in vitro activation and pre-analytical conditions [30]. Defining a rigorous laboratory methodology is essential to provide reliable and comparable results, especially in the setting of complex inflammatory situations like ECMO.

We sought to determine the effect of blood drawing and sample preparation on PLA measurement. Peripheral venous blood was obtained from seven healthy volunteers who were not taking any anti-platelet medication. Blood was drawn by clean venipuncture of a large antecubital vein, after applying a tourniquet, using a 21-gauge needle, and was collected into vacuum tubes containing 3.2% buffered sodium citrate or CTAD (sodium buffered citrate, theophylline, adenosine, and dipyridamole). Hirudin was not tested in this experiment because of its limited application in hemostasis laboratory, except for the whole-blood impedance aggregometry method. The first milliliters were discarded. The tubes were gently inverted once, immediately after blood collection. Blood was labeled using CD45-FITC and CD41-PE antibodies, with 15 min incubation time in darkness, without lysis or fixation. Samples were then diluted in phosphate buffer saline (final blood dilution 1:44) and analyzed, the events being acquired on Navios (Beckman Coulter) or Lyric (Becton Dickinson) (Figure 2). We evaluated (i) the type of anticoagulant solution, (ii) the delay between sampling and immunolabeling (15, 60, and 150 min), (iii) the sample stability after immunolabeling (0 and 90 min), and (iv) the effect of strong vortex agitation (10 s at 3200 rpm), prior to immunolabeling.

The results were as follows—(i) compared to sodium citrate, CTAD had a protective effect by reducing the rate of in vitro PLA formation prior to immunolabeling (*p* = 0.037) (Figure 2c); (ii) the time-interval between blood collection and immunolabeling had a marked effect on PLA levels; indeed, considering a mean level of 17% PLAs, an increase in PLAs by 1.7% was noticed every 10 min of delay with sodium citrate as anticoagulant (95% CI 1.1 to 2.3, *p* < 0.0001) and by 0.8% with CTAD (95% CI 0.2 to 1.4, *p* = 0.005); (iii) in vitro PLA formation still occurred after immunolabeling with an increase in PLA levels from baseline (0 min) to 90 min, using sodium citrate (15.7 ± 4.0% to 37.5 ± 7.9%; *p* = 0.0003) and CTAD (16.5 ± 5.6% to 42.0 ± 10.4%; *p* = 0.0004); (iv) finally, strong vortex agitation prior to immunolabeling had no significant effect on PLA levels, whatever the anticoagulant (*p* = 0.87). Using conditions that minimize in vitro PLA formation (15 min delay between blood collection and immunolabeling and no delay between immunolabeling and flow cytometric assay), mean PLA levels with sodium citrate and CTAD were, respectively, 17.8 ± 7.4% and 16.9 ± 6.2% (*p* = 0.7).

These results demonstrate that sample preparation before flow cytometry assay is critical, as it induces a significant variability in the PLA levels.

### 5.3. Optimal Conditions for Whole Blood Cytometric Measurement of PLAs: A Proposal

Our study was consistent with previous results published by Harding et al. regarding whole blood flow cytometric analysis of circulating platelet–monocyte aggregates (PMA) [46]. They reported a significant effect of anticoagulant type on PMA levels, in vitro PMA formation being better prevented with EDTA and sodium citrate than with heparin, hirudin, and PPACK. Indeed, platelet activation is likely best prevented by calcium chelators than with anticoagulant agents. Venipuncture led to lower PMA levels, compared to intravenous cannulas. Delay prior to immunolabelling induced significant in vitro PMA formation. Since red blood cell lysis could lead to platelet activation by ADP released from erythrocytes, immediate fixation should be performed along with red cell lysis, to avoid increase in PLA levels. Hence, Harding et al. demonstrated that erythrocyte lysis together with fixation after immunolabelling did not affect PMA and that PMA remained stable over 24 h when fixed and stored at 4 °C [46]. In line with these results, we found similar levels of total PLAs after dilution of the labeled blood samples with either phosphate buffer saline or FACS-lysing solution^®^.

Regarding the type of anticoagulant solution, CTAD seemed to offer the best protection from in vitro spontaneous PLA formation [44]. EDTA should be avoided as it could cause in vitro dissociation of platelets from leukocytes, notably through an αIIbβ3 integrin dissociation [44,47].

The fixation with paraformaldehyde along with erythrocyte lysis after immunolabeling seems to offer acceptable sample stability and reproducibility up to 24 h at 4 °C, without any effect on the PLA levels [43,44,46]. On the contrary, prefixation and red cell lysis before immunolabeling should not be used, as it can induce considerable change in the PLA formation dynamic, with an important increase in the PLA levels [41]. Additionally, since most PLAs are formed with monocytes, LPS-containing solutions should not be used.

Finally, due to a high number of platelets in the whole blood (10- to 100-fold higher than leukocytes), PLA analysis can be greatly altered by coincidence, i.e., double positivity occurring from non-interacting coinciding leukocytes and platelets or platelet-derived extracellular vesicles [48,49,50]. Coincidence is exacerbated by small dilution of samples and a high cytometric flow rate. Dilution and flow rates impact should be determined in each laboratory and should be stated in PLA cytometric protocols. Although final blood dilution should be above 1:40 to minimize coincident events detection [48], increasing blood dilution might induce PLA dissociation. Several approaches need to be considered to reduce coincident events measurement, including doublet-discriminator strategy [48,51] and imaging flow cytometry [52].

The optimized conditions we propose for whole blood cytometric measurement of PLAs, in order to minimize artefactual in vitro platelet activation and PLAs formation, are summarized in Table 2.

## 6. Conclusions

In the setting of ECMO, platelets could be involved in both bleeding and thrombotic complications. Indeed, ex-vivo studies showed either no effect or decreased platelet reactivity, which could be consecutive to vast in vivo activation of platelets in the ECMO circuit and, therefore, consumption and clearance of activated platelets and loss of reactivity. Upon activation, platelets might also interact with the inflammation system, resulting in pro-thrombotic effects. It is of utmost importance to better understand these mechanisms, in order to prevent and treat bleeding and thrombosis in the setting of ECMO. In this context, it would be valuable to study PLA formation to better assess the involvement of platelets in the inflammatory complications of ECMO. Moreover, PLAs could serve as markers of the modulation of thrombo-inflammation in patients receiving antithrombotic treatment in this setting of acute inflammation. Owing to the high reactivity of human platelets and the dynamic nature of platelet–leukocyte interaction, flow cytometric measurement of PLAs could be challenging and requires strictly defined pre-analytical conditions. International standardization is crucial to ensure comparability and reproducibility of future clinical trials.

## Figures and Tables

**Figure 1 jcm-09-02361-f001:**
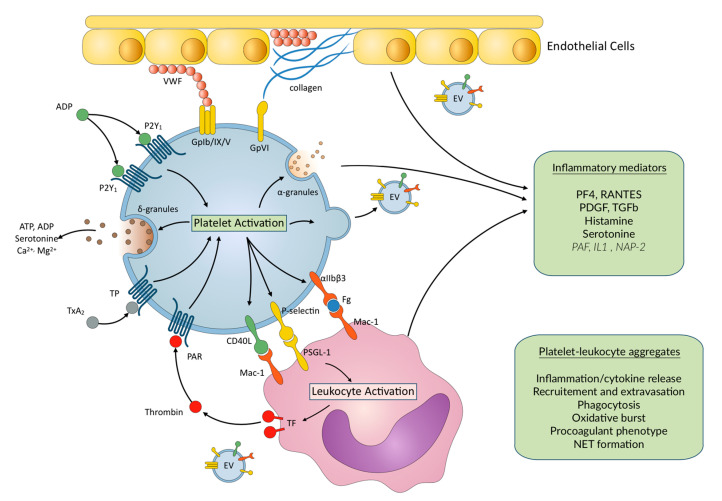
Main platelet–leukocyte interactions and inflammatory mediators released upon platelet activation. EC—endothelial cells; EV—extracellular vesicles; IL1—Interleukin 1; NAP-2—Neutrophil Activating Peptide-2; NET—neutrophil extracellular trap; PAF—platelet activating factor; PAR—protease activated receptor; PF4—platelet factor 4; PDGF—Platelet Derived Growth factor; TF—Tissue factor; TGFβ—Transforming Growth Factor β. VWF—von Willebrand factor. Mediators are stored in platelets. In italics—synthetized mediators/proteolytic cleavage of stored precursors upon platelet activation.

**Figure 2 jcm-09-02361-f002:**
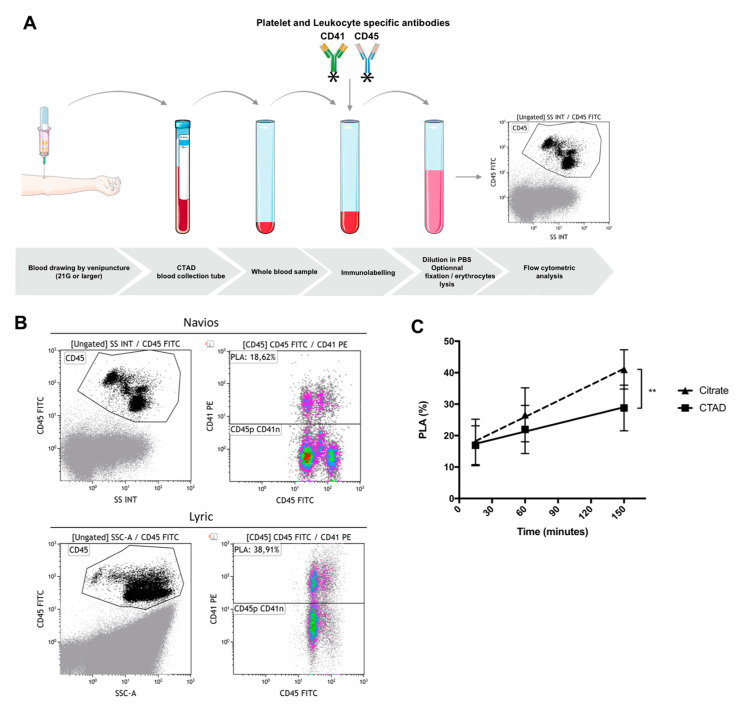
Platelet–leukocyte aggregates (PLAs) analyzed with flow cytometry (**A**). Schematic representation of sample preparation for PLA measurement in whole blood. Venous blood was drawn by puncture of a large vein using a 21-gauge needle and collected into tubes containing CTAD (sodium buffered citrate, theophylline, adenosine, and dipyridamole). The first milliliters were discarded. Blood sample was labeled using antibodies against CD45 and CD41 (20 µL of blood + 20 µL of PBS buffer + 10 µL of each antibody), with fixed incubation time (<1 h), followed by dilution (≥1/40) in PBS or with lysis and fixation buffer. (**B**) Example of samples acquired on Navios (top) or Lyric (bottom) flow cytometer. Blood was labeled using antibodies against CD45-FITC and CD41-PE, with 15 min of incubation (no lysis, no wash protocol). Final blood dilution was 1:44. Flow rate was at 70 events/sec and 10,000 events were acquired. Leukocytes (CD45) were defined as CD45pos side scatter high [CD45] (left panels). Then PLAs in [CD45] gated dot plots, were identified with the co-expression of CD45 and CD41 (right panels). (**C**) Effect of time prior to immunolabeling on PLAs. Blood was drawn into tubes containing 3.2% buffered sodium citrate or CTAD. Immunolabeling and processing of whole blood samples were performed 15, 60, and 150 min, following blood collection. Blood was labeled using CD45-FITC and CD41-PE antibodies (cf above), without lysis or fixation. Events were acquired on the Lyric system (Becton Dickinson). For every 10 min of time-interval prior to immunolabeling, PLA levels increased by 1.7% (95% CI 1.1 to 2.3, *p* < 0.0001) in citrate-anticoagulated blood (closed triangles, dashed line) and 0.8% (95% CI 0.2 to 1.4, *p* = 0.005) in CTAD-anticoagulated blood (closed squares, plain line). Linear regression analysis demonstrated a reduced rate of in vitro PLA formation, prior to immunolabeling with CTAD (solid line), compared to sodium citrate (dashed line) (*p* = 0.037). Data are reported as mean ± SEM (*n* = 7). SS INT and SSC-A—side scatter.

**Table 1 jcm-09-02361-t001:** Main results of platelet aggregation studies with Multiplate^®^.

Type of ECMO and PumpNumber of Patients Study Reference	Blood Collection	Results
Centrifugal pump Jostra pumphead, Maquet^®^ VA *n* = 7; VV *n* = 3 Nair et al. [12]	During ECMO. Citrated whole blood.	Patients with platelet count > 100 × 10^9^/L 50% to 72% of patients within low range of ADP, TRAP, collagen aggregation, and ristocetin agglutination
VA *n* = 26; VV *n* = 12 Tauber et al. [13]	Before, after 24 h and 48 h initiation of ECMO therapy and 24 h after ECMO termination. Hirudin anticoagulated whole blood.	30% to 40% decrease in aggregation with ADP, TRAP, AA after 24 h/baseline. After 48 h, aggregation with TRAP = baseline, ADP and AA lower/baseline Return to baseline after 24 h
ROTAFLOW^®^ or CARDIOHELP^®^ centrifugal pumps. VV *n* = 20 Wand et al. [14]	Before, 6 h, 1, 2, 3 and 7 days after the start of ECMO therapy. Whole blood sample on heparin-anticoagulated and calcium-balanced tubes.	Patients with platelet count > 70 × 10^9^/L. Reduced platelet aggregation with ADP, TRAP, AA 6 h after ECMO initiation/before, spontaneous recovery on day 2 with values exceeding baseline afterwards
VA *n* = 23; VV *n* = 10 Balle et al. [15]	Every day from day 1 to day 7 in 33 patients. Hirudin anticoagulated whole blood.	Platelet aggregation with ADP, TRAP, AA: lower compared to healthy volunteers from day 1 up to day 7 but similar when analyzed relative to platelet count

TRAP—thrombin receptor activating peptide-6. AA—arachidonic acid. ADP—adenosine diphosphate.

**Table 2 jcm-09-02361-t002:** Sample preparation and flow cytometric whole blood measurement of circulating platelet–leukocyte aggregates (PLAs).

Sample Handling and Preparation	
Anticoagulant in blood sampling tube	Preferably CTAD (other option: sodium citrate)
Storage temperature	Room temperature
Centrifugation before immunolabeling	None (whole blood protocol)
Prefixation before immunolabeling	None
Erythrocyte lysis before immunolabeling	None
Processing time before immunolabeling	Fixed time to allow comparative analysis As short as possible, preferably < 1 h after blood withdrawal
Fixation after immunolabeling	Optional, paraformaldehyde fixation (0.5 to 1%)
Final blood dilution	≥1:40
Erythrocyte lysis after immunolabeling	Optional
Processing time after immunolabeling	Without delay if no fixation 24 h at 4 °C, after fixation
Cytometric flow rates	Low to medium, determined in each laboratory
Immunolabeling	
Platelet-specific antibody	Preferably: CD41 Other option: CD61
Leukocyte-specific antibody	CD45
Leukocyte subtype-specific antibody	Neutrophils: CD66b
	Monocytes: CD14
	T-cells: CD3, CD4 and CD8
	B-cells: CD19
	NK-cells: CD56

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
