# Peer review of "Platelet Functions During Extracorporeal Membrane Oxygenation. Platelet–Leukocyte Aggregates Analyzed by Flow Cytometry as a Promising Tool to Monitor Platelet Activation"

_jcm, 2020, doi:10.3390/jcm9082361_

Round 1
Reviewer 1 Report
Extracorporal membrane oxygenation (ECMO) is an intensive care device to support patients with respiratory defects. Blood is pumped through a system of tubes and membranes, where it gets in contact with artificial surfaces. Patients on ECMO are thus at risk of an acquired bleeding disorder. While it is recognized that loss of high molecular weight von Willebrand factor polymers contributes to the bleeding phenotype, altered platelet function is less well studied.
The review article entitled "Platelet functions during extracorporeal membrane oxygenation: A review with a focus on circulating platelet-leukocyte aggregate count by flow cytometry" by Mansour and coworkers aims to summerize our current knowledge. Moreover, this article provides some primary data on the impact of preanalytic problems in respect to accurately measure platelet-leukocyte aggregates.
Major comments
This article is overall well written. Some minor language issues should be corrected (see below).
- l. 377: leukocyte-platelet and leukocyte-microvesicle aggregates can readily be distinguished when scatter properties are used in conjunction with CD41 on the platelet side in respect to the leukocyte marker. Platelets constitute a distinct population in FSC/SSC analysis; microvesicles are - as correctly stated by the authors - much smaller. Indeed, most flow cytometers require special settings to unambiguously identify microvesicular structures in respect to cell debris. The authors should clarify this.
- The title should be rephrased: What is a "circulating platelet-leukocyte aggregate count"? The count is not circulating.
-l. 456-466: The authors should emphasize that red blood cells contain a relevant concentration of ADP. Any type of red blood cell lysis together with resting platelets leads to platelet activation or enhances platelet activation and is thus not indicated.
- Table 1: please use "et al." instead of "et al"; patient numbers between Table 1 and text are inconsistent (i.e. for the study by Balle and coworkers". Please recheck numbers.
Minor comments
- Some phrases are redundant: l. 77: "bleeding can induce (sic!) thrombocytopenia, which thereafter increases bleeding, especially in the case of massive bleeding".
- l. 102: "In shear stress induced by ECMO, shear-induced shedding may not require..."
- l. 164/184: CD63 is a marker of lysosomal granules and is - in part - also expressed on dense granules. It is not a bona fide marker of dense granules, as CD62P is a marker for alpha granules.
- l. 175 "and a decrease expression of P-selectin". Please rephrase or correct.
- please check for consistent use of hyphens throughout the manuscript: va-ECMO / va ECMO". Do not use a hyphen in "in vitro" (l. 393) etc. Please check for spaces (Table1 vs. Table 1).
Author Response
We would like to thank the reviewers for their comments. This is a point-by point answer to the reviewer suggestions.
Reviewer 1
Extracorporal membrane oxygenation (ECMO) is an intensive care device to support patients with respiratory defects. Blood is pumped through a system of tubes and membranes, where it gets in contact with artificial surfaces. Patients on ECMO are thus at risk of an acquired bleeding disorder. While it is recognized that loss of high molecular weight von Willebrand factor polymers contributes to the bleeding phenotype, altered platelet function is less well studied.
The review article entitled "Platelet functions during extracorporeal membrane oxygenation: A review with a focus on circulating platelet-leukocyte aggregate count by flow cytometry" by Mansour and coworkers aims to summerize our current knowledge. Moreover, this article provides some primary data on the impact of preanalytic problems in respect to accurately measure platelet-leukocyte aggregates.
Major comments
This article is overall well written. Some minor language issues should be corrected (see below).
- l. 377: leukocyte-platelet and leukocyte-microvesicle aggregates can readily be distinguished when scatter properties are used in conjunction with CD41 on the platelet side in respect to the leukocyte marker. Platelets constitute a distinct population in FSC/SSC analysis; microvesicles are - as correctly stated by the authors - much smaller. Indeed, most flow cytometers require special settings to unambiguously identify microvesicular structures in respect to cell debris. The authors should clarify this.
Our aim was not to quantify microvesicules, as we gated on the leukocyte population (CD45 positive/SSC). However, we cannot exclude that platelet microvesicles CD41+ bound to leukocytes were quantified along with PLAs, which is not a concern as platelet MVs also result from activation processes
- The title should be rephrased: What is a "circulating platelet-leukocyte aggregate count"? The count is not circulating.
The title has been modified:
"Platelet functions during extracorporeal membrane oxygenation. Platelet-leukocyte aggregates analyzed by flow cytometry as a promising tool to monitor platelet activation"
-l. 456-466: The authors should emphasize that red blood cells contain a relevant concentration of ADP. Any type of red blood cell lysis together with resting platelets leads to platelet activation or enhances platelet activation and is thus not indicated.
We do agree that red blood cell lysis could result in platelet activation and subsequent increase in PLA formation. However, Harding et al. (Thromb Haemost, 2007) showed that erythrocyte lysis using FACS-lysing solution® (Becton Dickinson) after incubation of blood samples with monoclonal antibodies, had no effect on platelet-monocyte aggregate measurements. In line with these results, we found similar levels of total PLAs after dilution of labelled blood samples with either phosphate buffer saline or FACS-lysing solution®. The use of a solution allowing fixation along with erythrocyte lysis would likely prevent platelet activation by ADP released from erythrocytes, and thus avoid in vitro formation of PLAs.
Two sentences have been added to address this issue, L-512 and 516: " Since red blood cell lysis could lead to platelet activation by ADP released from erythrocytes, immediate fixation should be performed along with red cell lysis to avoid increase in PLA levels."
" In line with these results, we found similar levels of total PLAs after dilution of labelled blood samples with either phosphate buffer saline or FACS-lysing solution® (data not shown).
- Table 1: please use "et al." instead of "et al"; patient numbers between Table 1 and text are inconsistent (i.e. for the study by Balle and coworkers". Please recheck numbers.
The changes have been made and the numbers corrected accordingly.
Minor comments
- Some phrases are redundant: l. 77: "bleeding can induce (sic!) thrombocytopenia, which thereafter increases bleeding, especially in the case of massive bleeding".
We do agree and the sentence has been deleted.
- l. 102: "In shear stress induced by ECMO, shear-induced shedding may not require..."
The sentence now reads: "In ECMO, shear-induced shedding…."
- l. 164/184: CD63 is a marker of lysosomal granules and is - in part - also expressed on dense granules. It is not a bona fide marker of dense granules, as CD62P is a marker for alpha granules.
We do agree that CD63 is not a specific marker of dense granules. The sentence now reads, L164: "Platelet activation was also measured by flow cytometry as the increase in expression of platelet surface P-selectin (CD62P, alpha granule membrane protein) and CD63 (lysosome and dense granule membrane protein) following activation, as well as bound fibrinogen to αIIbβ3..."
- l. 175 "and a decrease expression of P-selectin". Please rephrase or correct.
The correction has been made
- please check for consistent use of hyphens throughout the manuscript: va-ECMO / va ECMO". Do not use a hyphen in "in vitro" (l. 393) etc. Please check for spaces (Table1 vs. Table 1).
The changes have been made
Reviewer 2
The Review article by Mansour et al entitled “Platelet functions during extracorporeal membrane oxygenation: A review with a focus on circulating platelet-leukocyte aggregate count by flow cytometry” discusses platelet function and platelet-leukocyte aggregation (PLA) during extracorporeal membrane oxygenation (ECMO).
Overall the review is well written and describes many details of platelet function in ECMO patients and highlights the paucity of studies and inconsistencies reported in the literature. The authors also provide data regarding methodology for the determination of PLA in whole blood by flow cytometry.
Main comments.
The overall message of the review needs to be strengthened. The current title implies that the review will focus on circulating PLA in ECMO. However, only sparse information is provided about PLA in ECMO. This is fine, given the absence of studies examining PLA specifically in ECMO patients. In my opinion, the title should be changed to more accurately reflect the topic of the review, i.e. just platelet function during ECMO.
We do agree with the reviewer that we do not provide any data on PLAs in ECMO due to the absence of studies. However, since the potential use of PLAs in this setting as well as the methodological considerations are addressed; it seems important to us that flow cytometry analysis appears in the title.
We propose to change the title as follows:
"Platelet functions during extracorporeal membrane oxygenation.
Platelet-leukocyte aggregates analyzed by flow cytometry as a promising tool to monitor platelet activation."
The subheading in lines 390 and 391 “Methodological and pre-analytical considerations for whole blood measurement of PLAs under ECMO” implies that this methodology is specifically designed for ECMO, but in fact this is a general method to detect PLA in whole blood. If this method is proposed as a standard for PLA determination a more systematic approach is required. Statistically, are seven volunteers sufficient for this study? Data for leukocyte subtypes stained with CD66B, CD14, CD19, CD56 and CD3, CD4 and CD8 should also be shown. Were platelet doublets excluded?
As suggested by the reviewer, we have modified some titles (L-246 and 260) and added some subheadings:
- Flow cytometric whole blood measurement of platelet-leukocyte aggregates
General considerations (L-402)
Pre-analytical requirements: choice of anticoagulant, delay between sampling and immunolabelling, sample stability after immunolabelling, effect of strong vortex agitation prior to immunolabelling. (L-438)
Optimal conditions for whole blood cytometric measurement of PLAs: a proposal (L-505)
The number of healthy volunteers (n=7) we used was considered sufficient to test the different pre-analytical conditions, especially since the results were very consistent across the volunteers. However, we agree that a higher number of samples would be required to precisely define the reference range for our experimental setting.
We did not mean to specifically study the different subtypes of PLAs. The aim of our experiments was to assess the total PLAs levels using the pan-leukocyte antibody anti-CD45. Others have nicely addressed this issue (Finsterbusch, M et al. ; Gerrits, A.J et al.).
To help the reader it might be better to include additional subheadings since each subheading includes a lot of information.
Cf supra
Minor comments: changes have been made according to the reviewer suggestions
In abstract (line 31), please change “has been inconstantly…” to “has been inconsistently…”. Similarly, in other parts of the MS such as in line 200.
Table 1 describes the results of the studies presented. There is no need for extensive description of the same studies in the text (page 4).
Consistency: either aVWF or AVWF
Reviewer 2 Report
The Review article by Mansour et al entitled “Platelet functions during extracorporeal membrane oxygenation: A review with a focus on circulating platelet-leukocyte aggregate count by flow cytometry” discusses platelet function and platelet-leukocyte aggregation (PLA) during extracorporeal membrane oxygenation (ECMO).
Overall the review is well written and describes many details of platelet function in ECMO patients and highlights the paucity of studies and inconsistencies reported in the literature. The authors also provide data regarding methodology for the determination of PLA in whole blood by flow cytometry.
Main comments.
The overall message of the review needs to be strengthened. The current title implies that the review will focus on circulating PLA in ECMO. However, only sparse information is provided about PLA in ECMO. This is fine, given the absence of studies examining PLA specifically in ECMO patients. In my opinion, the title should be changed to more accurately reflect the topic of the review, i.e. just platelet function during ECMO.
The subheading in lines 390 and 391 “Methodological and pre-analytical considerations for whole blood measurement of PLAs under ECMO” implies that this methodology is specifically designed for ECMO, but in fact this is a general method to detect PLA in whole blood. If this method is proposed as a standard for PLA determination a more systematic approach is required. Statistically, are seven volunteers sufficient for this study? Data for leukocyte subtypes stained with CD66B, CD14, CD19, CD56 and CD3, CD4 and CD8 should also be shown. Were platelet doublets excluded?
To help the reader it might be better to include additional subheadings since each subheading includes a lot of information.
Minor comments.
In abstract (line 31), please change “has been inconstantly…” to “has been inconsistently…”. Similarly, in other parts of the MS such as in line 200.
Table 1 describes the results of the studies presented. There is no need for extensive description of the same studies in the text (page 4).
Consistency: either aVWF or AVWF
Author Response

(The authors gave the same response as above.)

Round 2
Reviewer 2 Report
I have reviewed the revised article by Mansour et al. I'm satisfied with the authors' replies to the reviewers' questions and have no additional comments.